# Feasibility of Ultrasound Attenuation Imaging for Assessing Pediatric Hepatic Steatosis

**DOI:** 10.3390/biology11071087

**Published:** 2022-07-20

**Authors:** Kyungchul Song, Nak-Hoon Son, Dong Ryul Chang, Hyun Wook Chae, Hyun Joo Shin

**Affiliations:** 1Department of Pediatrics, Severance Children’s Hospital, Endocrine Research Institute, Yonsei University College of Medicine, Seoul 03722, Korea; endosong@yuhs.ac (K.S.); hopechae@yuhs.ac (H.W.C.); 2Department of Statistics, Keimyung University, Daegu 42601, Korea; nhson@ms.kmu.ac.kr; 3Department of Radiology, Research Institute of Radiological Science and Center for Clinical Imaging Data Science, Yongin Severance Hospital, Yonsei University College of Medicine, Yongin-si 16995, Korea; rarara419@yuhs.ac

**Keywords:** ultrasonography, liver steatosis, attenuation imaging, fat quantification, children

## Abstract

**Simple Summary:**

Hepatic steatosis is associated with cardiovascular disease, diabetes mellitus, and liver cirrhosis. The increasing prevalence of hepatic steatosis among children has become a public health concern. Although liver biopsy is the gold standard for the diagnosis of hepatic steatosis, it has limited value because of invasiveness. Among imaging studies, ultrasonography is readily accessible and can be used to exclude other pathology, but its accuracy is limited by low sensitivity and specificity. Although magnetic resonance imaging is highly accurate for liver steatosis and fibrosis, its application is curbed in children as it requires sedation and longer scan time. Recently, attenuation imaging (ATI) has emerged as a new modality for quantifying fat deposition in the liver in real time. However, investigations of ATI for pediatric hepatic steatosis are still in their preliminary stages. Thus, we investigated the feasibility of ultrasound attenuation imaging (ATI) for assessing pediatric hepatic steatosis. This study demonstrated that ATI can differentiate fatty liver from normal liver as well as moderate to severe fatty liver from mild fatty liver. Thus, ATI may be useful for identifying children who require liver biopsy and early treatment.

**Abstract:**

We investigated the feasibility of ultrasound attenuation imaging (ATI) for assessing pediatric hepatic steatosis. A total of 111 children and adolescents who underwent liver ultrasonography with ATI for suspected hepatic steatosis were included. Participants were classified into the normal, mild, or moderate–severe fatty liver group according to grayscale US findings. Associations between clinical factors, magnetic resonance imaging proton density fat fraction, steatosis stage and ATI values were evaluated. To determine the cutoff values of ATI for staging hepatic steatosis, areas under the curve (AUCs) were analyzed. Factors that could cause measurement failure with ATI were assessed. Of 111 participants, 88 had successful measurement results. Median ATI values were significantly increased according to steatosis stage (*p* < 0.001). Body mass index (BMI) was a significant factor for increased ATI values (*p* = 0.047). To differentiate fatty liver from normal liver, a cutoff value of 0.59 dB/cm/MHz could be used with an AUC value of 0.853. To differentiate moderate to severe fatty liver from mild fatty liver, a cutoff value of 0.69 dB/cm/MHz could be used with an AUC value up to 0.91. ATI can be used in children as an effective ultrasonography technique for quantifying and staging pediatric hepatic steatosis.

## 1. Introduction

Non-alcoholic fatty liver disease (NAFLD) is a chronic disease induced by excessive fat accumulation in the liver that includes simple steatosis, steatohepatitis, fibrosis, and cirrhosis [1,2]. As NAFLD is associated with cardio-metabolic risk factors, it has been additionally related to all-cause deaths due to cardiovascular disease and diabetes mellitus as well as liver cirrhosis [3,4,5]. The prevalence of NAFLD among youths increased from 19.34 million in 1990 to 29.49 million in 2017 according to the Global Burden of Disease database-based study, and corresponding values increased 2.7-fold from the 1980s to 2007–2010 in the USA [6,7]. In Korea, the prevalence of NAFLD increased from 8.17% in 2009 to 12.05% in 2018 among children and adolescents [8].

As NAFLD is often asymptomatic and reversible with treatment in its initial stages, an early diagnosis is important to prevent severe complications [1,9]. The gold standard for the diagnosis of NAFLD is liver biopsy, but it has limited value because of its invasiveness, the small size of the samples collected and the non-uniformity of the disease itself [1,10]. Although the alanine aminotransferase (ALT) test is recommended for screening NAFLD, serum ALT can be elevated in other liver diseases and NAFLD can be detected even in children with normal ALT [11]. Among imaging studies, ultrasonography (US) is readily accessible and can be used to exclude other pathology, but its accuracy is limited by low sensitivity and specificity [1,12]. Although MRI is highly accurate for liver steatosis and fibrosis, its application is curbed in children as it requires sedation and longer scan time [9].

Based on evidence that showed increased attenuation of the US beam in hepatic steatosis, the controlled attenuation parameter (CAP) was a widely accepted imaging modality used to quantify liver fat [12,13]. However, the clinical application of CAP was restricted by frequent measurement failure and non-available anatomic imaging [13]. In response, attenuation imaging (ATI) has emerged as a new modality for quantifying fat deposition in the liver using real-time US [12,14]. Recent studies have reported high correlations between ATI and the MRI proton density fat fraction (PDFF), and ATI has shown excellent performance for detecting hepatic steatosis in adults [12,15]. Gatos et al. reported that liver ultrasound attenuation helped to accurately grade hepatic steatosis in adults with NAFLD [16]. Jeon et al. reported that 2-D ultrasound ATI was useful when assessing hepatic steatosis in adults with chronic liver disease with MRI-PDFF as the reference standard [15]. Ferraioli et al. recently summarized cutoff values for diagnosing hepatic steatosis in adults with liver biopsy as a reference standard [17]. However, investigations of ATI for pediatric hepatic steatosis are still in their preliminary stages [18]. Because the early detection and correction of hepatic steatosis is important in the pediatric population, validating the use of ATI for the accurate quantification of hepatic steatosis is of clinical necessity in children.

Therefore, the purpose of this study was to demonstrate the feasibility of ATI for assessing hepatic steatosis in children and to determine cutoff values for ATI to achieve acceptable diagnostic performance for staging hepatic steatosis.

## 2. Materials and Methods

### 2.1. Subjects

Children and adolescents (≤18 years old) who underwent liver US with ATI from March to December 2021 according to clinical demand to evaluate liver disease due to obesity or increased liver function test results were included. We separated participants who failed to attain reliable ATI values. Measurement failure was defined when any of three criteria was met: (1) R^2^ value < 0.8, (2) interquartile range (IQR)/median of ATI value > 0.3, or (3) no visible homogeneous color map for more than half of the intended region-of-interest (ROI). These criteria were from the vendor’s provided methods for adults, because there are no criteria applicable for children yet. The R^2^ value represents the coefficient of determination and R^2^ ≥ 0.8 is considered to indicate a valid measurement from an appropriate ROI location. The IQR/median value was used to evaluate the reliability of the measurements [14,15].

Participants with successful ATI measurements were then classified into the normal, mild fatty liver and moderate–severe fatty liver group according to liver parenchymal echogenicity on US, which is a classification method generally used to evaluate hepatic steatosis [17,18,19]. The mild fatty liver group included participants whose livers showed brighter echogenicity compared with the renal cortex on US [19]. The moderate to severe fatty liver group included participants whose portal vein wall was blurred from increased liver parenchymal echogenicity and who had diaphragms that were poorly visualized from US beam attenuation [19]. We classified participants according to liver echogenicity and not by BMI, because liver parenchymal conditions can differ from BMI, and we considered liver echogenicity as a more direct representation of the parenchymal condition.

Age, sex, height, weight, body mass index (BMI), systolic blood pressure (SBP), diastolic blood pressure (DBP), aspartate transaminase (AST), and ALT values obtained during the outpatient clinic visit within 7 days of the US examination were reviewed retrospectively using electronic medical records.

### 2.2. MRI Acquisition

When participants underwent liver MRI including PDFF within one month of the US examination, PDFF values were also included in the analyses. Abbreviated liver MRI was performed according to clinical demand when participants were suspected of fatty liver after the initial US examination and when clinicians wished to know the quantitative fat amount. MRI was performed in a 3T system (Ingenia Elition X, Philips Medical Systems, Veenpluis, Best, The Netherlands) using a pediatric body coil. Included sequences were (1) axial and coronal single-shot fast spin echo T2 weighted images for anatomic imaging, (2) axial 3D volumetric multi-echo gradient sequence for PDFF and T2* decay quantification, and (3) axial gradient echo MR elastography (MRE) for liver fibrosis quantification. The MRI parameters for PDFF were as follows; repetition time (TR) 5.7 ms, echo time (TE) 2.6 ms, matrix 160 × 160, slice thickness 6 mm, flip angle 3°, number of signal averages 1, with six gradient echoes from 0.9 to 4.4 ms. The total acquisition time was 15 s. In addition, the MRI parameters for MRE were as follows; TR 1000 ms, TE 60 ms, matrix 64 × 64, slice thickness 10 mm, flip angle 90°, number of signal averages 2, with a total acquisition time of 24 s. The chosen MRE techniques were the same as those used in previous studies in our institution. Only participants who could cooperate without sedation underwent MRI.

After the acquisition, the maximum longitudinal diameter of the right liver lobe (cm), PDFF (%), T2* (ms) value, and MRE (kPa) value of the liver were collected. To obtain PDFF, T2*, and MRE values, four ROIs were drawn in the liver avoiding hepatic vessels and artifacts using the axial slices of each sequence by an experienced pediatric radiologist and the mean value of the four measurements was used as a representative value (Figure 1 and Figure 2). Steatosis grades could be also categorized using cutoff values of PDFF identified in a previous study; S1 (mild) for PDFF > 6%~<17.5%, S2 (moderate) for PDFF ≥ 17.5%~<23.3%, and S3 (severe) for PDFF ≥ 23.3% [20].

### 2.3. ATI Acquisition

Liver US with ATI was performed by a pediatric radiologist with 12 years of experience using a C1-8MHz convex transducer (PVI-475BX) of the Aplio i800 (Canon Medical Systems, Otawara, Japan) machine. ATI can measure fat deposition in the liver by quantifying attenuation coefficient in fatty liver. The higher the attenuation, the more US echo decreases according to depth, resulting in a steeper slope. ATI can quantify this slope, otherwise known as the attenuation coefficient [21]. For abdominal US evaluation including ATI, participants were in the supine position with their right arm elevated. They were asked to fast for 4 h before the examination. Participants were in shallow breathing or stable free breathing to avoid forced breath-holds or artifacts from poor cooperation. The transducer was placed perpendicular to the skin in the right intercostal space without compression to avoid shadowing or artifacts. After activating the ATI mode during grayscale US, a large fan-shaped color-coded sampling box automatically appeared in the liver parenchyma. Smaller fan-shaped ROIs approximately 2 × 4 cm in size were placed to include adequate liver parenchyma when color seemed homogeneous with a R^2^ value equal to or more than 0.8 (Figure 1 and Figure 2). A total of five measurements were repeated at separate intervals and the median values of ATI, IQR and IQR/median values were obtained. In addition to ATI, shear-wave elastography (SWE) and shear-wave dispersion (SWD) were measured, and median values were chosen as representative values after ten repeated measurements.

### 2.4. Statistical Analysis

Statistical analyses were performed using SPSS version 26.0 (IBM Corp., Armonk, NY, USA). Variables followed normal distribution with *p*-values > 0.05 for the Kolmogorov–Smirnov test. Participant demographics, MRI and US results were compared among the three fatty liver groups using one-way analysis of variance (ANOVA) with a subgroup analysis being performed using Bonferroni correction to suppress the increase in type I errors. The Chi-square test was used for categorical variables and the independent two-sample *t*-test was used for MRI values. To identify significant factors for fatty liver, ordinal logistic regression analyses were performed. In addition, a linear regression analysis was performed to identify significant factors that could predict increased ATI values in the liver. To identify cutoff values and diagnostic performance of ATI for predicting fatty liver according to US and PDFF, areas under the curve (AUCs) were analyzed. An optimal cutoff value was chosen for each parameter to maximize the sum of sensitivity and specificity in AUC analysis. Furthermore, we analyzed factors that could cause measurement failure with ATI using the independent *t*-test, Chi-square test and logistic regression test. P-values less than 0.05 were considered statistically significant.

## 3. Results

### 3.1. Demographics

During the study period, 111 pediatric participants underwent liver US with ATI and 23 participants had measurement failure. A total of 88 participants (79.3%, mean 10.4 ± 2.6 years old, age range 6–17.8 years, M:F = 47:41) had successful results and they were classified into the normal (*n* = 31), mild fatty liver (*n* = 30) and moderate–severe fatty liver (*n* = 27) groups according to grayscale US. With only a relatively small number of participants with moderate or severe fatty liver, participants with moderate fatty liver and those with severe fatty liver could not be analyzed as separate groups. Age, BMI, SBP, AST and ALT were significantly different between the three groups (Table 1). Height, weight, and BMI progressively increased across fatty liver grade. Therefore, we included age as a variable in the regression analysis to adjust for age difference.

Among 88 participants, only 20 participants (22.7%, mean 11.1 ± 1.9 years old) underwent liver MRI with PDFF (mean 22.1 ± 11.7 %, range 6.1–43.4%). Based on the grayscale US findings, five participants were included in the mild fatty liver group, while the remaining 15 participants were included in the moderate–severe fatty liver group. None had normal liver parenchymal echogenicity on US. For MRI measurements, PDFF was significantly higher in the moderate–severe fatty liver group compared with the mild fatty liver group (mean 25.2% vs. 12.9 %, *p* = 0.04).

Among the US measurements, median ATI values significantly and gradually increased as the severity of steatosis increased (0.54, 0.63, and 0.73 dB/cm/MHz, respectively, *p* < 0.001) (Figure 3). SWE values significantly increased in the fatty liver group compared with the normal group (1.26–1.29 m/s vs. 1.19 m/s, *p* = 0.019–0.026).

### 3.2. Results of the Ordinal Logistic Regression Test for the Fatty Liver Groups

On the univariate ordinal logistic regression test, age, sex, height, weight, BMI, SBP, AST, and ALT were significant variables for predicting the three fatty liver groups determined by US echogenicity (Table 2). Among the US measurements, median ATI values (odds ratio [OR] 15.536, 95% confidence interval [CI] 10.186–20.886, *p* < 0.001) and SWE values (OR 3.746, 95% CI 0.511–6.980, *p* = 0.023) were significant factors for grading fatty liver. Of these variables, only BMI (OR 0.453, *p* = 0.197–0.709, *p* = 0.001) and ATI values (OR 22.09, 95% CI 11.949–32.23, *p* < 0.001) remained significant after the multivariate analysis

### 3.3. Results of the Linear Regression Test for ATI Values

On univariate linear regression analysis, weight, BMI, SBP, AST, ALT, PDFF and T2* values of MRI were significant factors for increased ATI values (Table 3). We included those variables except for AST which was excluded due to multicollinearity (variance inflation factor [VIF] 20.116 for AST and 19.836 for ALT), in the multivariate linear regression analysis. As a result, BMI was found to be the only significant factor for increased ATI values (*p* = 0.047).

### 3.4. Diagnostic Performance of ATI Values

After AUC analysis, using a cutoff value of 0.59 dB/cm/MHz, ATI showed an AUC value of 0.853 (95% CI 0.762–0.920) with a sensitivity of 80.7% and specificity of 77.4% for differentiating fatty liver (mild, moderate, severe) from the normal group according to the US criteria (Table 4, Figure 4a). When differentiating moderate to severe fatty liver from mild fatty liver, a cutoff value of 0.69 dB/cm/MHz showed an AUC value of 0.778 (95% CI 0.649–0.878) with a sensitivity of 70.4% and specificity of 83.3% (Figure 4b).

When hepatic steatosis was graded using published cutoff values for PDFF, 9 participants had S1 (median PDFF 12%, range 6.1–17%) and 11 participants had S2–3 (median PDFF 29%, range 20.3–43.4%). Participants with S1 had a median ATI value of 0.6 dB/cm/MHz ranging 0.59–0.74 dB/cm/MHz, while participants with S2–3 had a median ATI value of 0.74 dB/cm/MHz ranging 0.65–0.98 dB/cm/MHz. After the AUC analysis, an ATI cutoff value of 0.69 dB/cm/MHz could effectively differentiate S2–3 from S1 with an AUC of 0.909 (95% CI 0.695–0.990) (Figure 4c). This cutoff value was identical to the cutoff value obtained for differentiating moderate to severe fatty liver according to the grayscale US criteria (Table 4).

### 3.5. Factors Related to Measurement Failure with ATI in Pediatric Participants

Measurement failure occurred in 23 of the 111 pediatric participants and participants with measurement failure were of younger age (mean 8.3 ± 1.4 vs. 10.0 ± 2.6 years old, *p* < 0.001), decreased weight (36.4 ± 8.3 vs. 53.8 ± 18.4 kg, *p* < 0.001), decreased BMI (19.9 ± 3.6 vs. 24.6 ± 4.2 kg/m^2^, *p* < 0.001) and normal liver echo on US (78.3% vs. 35.3%, *p* < 0.001), compared with participants with successful measurements (Table 5). On the univariate logistic regression test, age, sex, BMI and normal liver echo on US were all significant variables associated with measurement failure, and only BMI showed significant association on the multivariate analysis.

## 4. Discussion

ATI has been suggested as an answer to an increasing need for accurate fat quantification in the liver, and shows advantages over other techniques as ATI values can be simply obtained with grayscale US examinations [22]. With ATI, radiologists measure the degree of US beam attenuation according to depth by using a rudimentary echo decay profile after removing systemic influences on US [23]. Operators can easily measure ATI values during grayscale US through the automatic removal of vessels and artifacts. ATI has shown better reproducibility in its results, lower measurement failure, and comparable diagnostic performance compared to CAP in adult studies [24,25]. In adults, the AUC values of ATI for staging steatosis grades 1–3 have been reported to range from 0.76 to 0.99 [23]. In addition, the quantification of hepatic steatosis using ATI is not affected by the degree of liver fibrosis [26].

Our study is meaningful because ATI has not been assessed extensively in the pediatric population. In comparison to the broad research already done in adults, pediatric studies on ATI have been limited to healthy children [18]. Cailloce et al. demonstrated that the median ATI value was 0.65 dB/cm/MHz with a range of 0.5–0.81 dB/cm/MHz in 77 healthy children [18]. In their study, 89.5% of the children had valid measurements [18]. They suggested that a thickened hepatocyte layer pattern in younger children could explain the increased US beam attenuation seen on ATI and be the reason for the slightly increased median ATI value compared to previously published values in adults (0.31–0.75 dB/cm/MHz) [18]. In our study, 79.3% of participants had successful ATI measurements. The median ATI value in the normal group was 0.54 ± 0.09 dB/cm/MHz and it was comparable with past studies. One recent study demonstrated that CAP showed better performance than grayscale US for diagnosing hepatic steatosis in a pediatric population but there is no established data for ATI in diseased children [27].

Using MRI-PDFF as reference, the known cutoff value for diagnosing hepatic steatosis in adults is 0.59 dB/cm/MHz for ATI with an AUC of 0.76, sensitivity of 88%, and specificity 62.2% [15]. In addition, for discriminating S2–3 from S0–1, a cutoff value of 0.72 dB/cm/MHz was used for adults with an AUC of 0.95, sensitivity of 100%, and specificity of 78% [17,28]. These results were comparable to our study which showed cutoff values of 0.59 and 0.69 dB/cm/MHz, with AUC values of 0.853 and 0.909, respectively. To our knowledge, our study is the first to demonstrate a specific cutoff value for ATI using US and MRI as references for pediatric hepatic steatosis.

In our retrospective study, 20.7% of the examinations ended with measurement failure. This was a relatively higher rate of failure compared to adults and possibly because we applied a strict definition for measurement failure. Decreased R2 values were frequently encountered during examination and this tendency also needs be further validated in younger participants. In addition, we repeated measurements for the ATI values five times separately. Obtaining stable results during longer scan times with repeated measurements is not easy for children and steady breath holding is also difficult. This might have affected the number of valid measurements in our study population. Apart from the measurement technique, there was also a tendency for increased measurement failure in the normal liver and younger children with lower BMI. In healthy younger children, decreased liver volume and a smaller measurable area in the liver parenchyma without vessels could be the reason for fewer valid measurements. Further research on the reliability of ATI measurements and the definition of measurement failure in the pediatric population is warranted.

Additionally, we performed SWE and SWD during US acquisition. SWE is used to quantify liver fibrosis and SWD is related to tissue viscosity that can be seen in liver inflammation [18,29]. These are new techniques, especially SWD, and expected to quantify pathophysiologic characteristics of the liver independently [29]. In our study population, SWE was significantly increased in the fatty liver group compared to the normal group, even though SWE and SWD did not show significant results in regression tests. This might be because our study cohort included a relatively small number of participants in the moderate to severe fatty liver group and participants with liver fibrosis accompanied by steatosis. Further studies with a larger population are needed to demonstrate the clinical implications of SWD in pediatric NAFLD.

Our study has several limitations. First, we did not obtain a histologic confirmation of hepatic steatosis in children. Instead, we assessed fatty liver and steatosis degree with the generally acknowledged method of using grayscale US and the most accurate method of using MRI-PDFF. Using these methods, we found cutoff values that were not very different from those reported in other studies. Second, one radiologist performed the US examinations and inter- and intra-observer variability was not assessed in this retrospective study. However, the performing radiologist was thought to have adequate experience with pediatric US including shear-wave elastography and we believe that the results were consistent and trustworthy. In addition, another study demonstrated that ATI had high intra- and inter-observer reproducibility [30]. Third, a relatively small number of participants were included in our study. Only a few of the participants included in our study population had moderate to severe fatty liver, and we could not classify them further. In addition, children with normal liver echogenicity on grayscale US did not undergo MRI and MRI results were only available for participants with fatty liver. Substantial number of the participants did not undergo MRI due to matter of cost and time in this retrospective study. Thus, accuracy of validation is limited due to a small number of the participants who underwent MRI. However, we validated the feasibility of ATI through comparing the ATI cutoff value for predicting mild fatty liver and moderate to severe fatty liver graded by gray scale US and those by MRI. A larger study population is needed to validate the ATI technique for pediatric hepatic steatosis.

## 5. Conclusions

Our study showed that successful ATI measurement was possible in approximately 80% of children and ATI values significantly increased with the degree of fatty liver and BMI values. Using cutoff values of 0.59 and 0.69 dB/cm/MHz, ATI could effectively discriminate the presence of fatty liver and differentiate moderate to severe fatty liver in children, respectively. ATI has the potential to be an effective US technique for quantifying pediatric hepatic steatosis, but measurement methods need to be further validated to assure reliable results in children.

## Figures and Tables

**Figure 1 biology-11-01087-f001:**
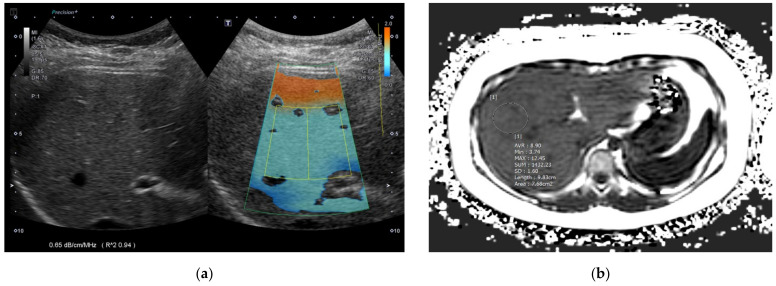
A 7-year-old girl in the mild fatty liver group according to grayscale US. (**a**) The ATI value was 0.65 dB/cm/MHz and (**b**) MRI-PDFF value was 8.9% (S1: PDFF > 6%). US: ultrasonography; ATI: attenuation imaging; MRI: magnetic resonance imaging; PDFF: proton density fat fraction.

**Figure 2 biology-11-01087-f002:**
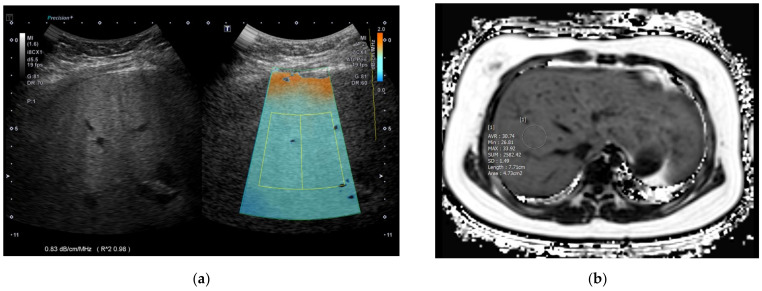
An 11-year-old girl in the moderate to severe fatty liver group according to grayscale US. (**a**) The ATI value was 0.83 dB/cm/MHz and (**b**) MRI-PDFF value was 30.7% (S3: PDFF > 23.3%). US: ultrasonography; ATI: attenuation imaging; MRI: magnetic resonance imaging; PDFF: proton density fat fraction.

**Figure 3 biology-11-01087-f003:**
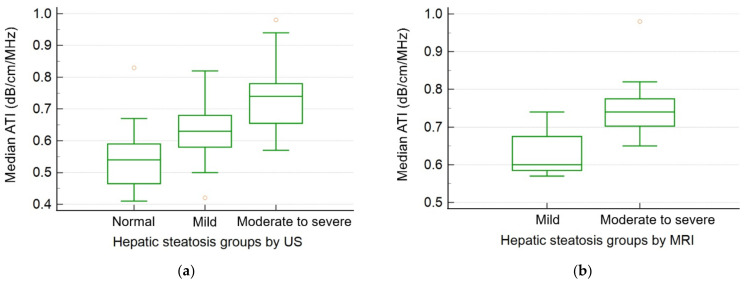
Box–whisker plots of median ATI values for each hepatic steatosis group (**a**) according to US echogenicity and (**b**) MRI-PDFF value. US: ultrasonography; ATI: attenuation imaging; MRI: magnetic resonance imaging; PDFF: proton density fat fraction.

**Figure 4 biology-11-01087-f004:**
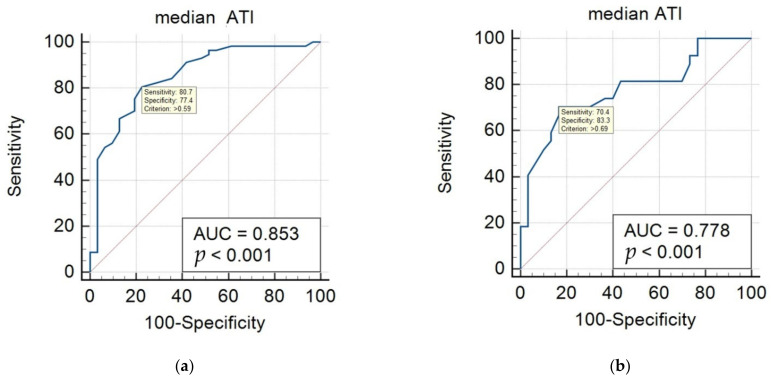
ROC curves of median ATI values for staging hepatic steatosis according to the grayscale US and MRI-PDFF. (**a**) For differentiating fatty liver (mild, moderate, severe) from the normal group according to the US criteria, a cutoff value of 0.59 dB/cm/MHz showed an AUC value of 0.853 (95% CI 0.762–0.920). (**b**) For differentiating moderate to severe fatty liver from mild fatty liver according to the US criteria, a cutoff value of 0.69 dB/cm/MHz showed an AUC value of 0.778 (95% CI 0.649–0.878). (**c**) For differentiating moderate to severe fatty liver from mild fatty liver according to MRI-PDFF, a cutoff value of 0.69 dB/cm/MHz showed an AUC value of 0.909 (95% CI 0.695–0.990). US: ultrasonography; ATI: attenuation imaging; MRI: magnetic resonance imaging; PDFF: proton density fat fraction; AUC: area under the curve; CI: confidence interval.

**Table 1 biology-11-01087-t001:** Comparison of participant demographics and MRI and US measurements among the three fatty liver groups (normal, mild, and moderate–severe) according to grayscale US.

Variables		Normal Liver Group ^1^ (*n* = 31)	Mild Fatty Liver Group ^2^ (*n* = 30)	Moderate–Severe Fatty Liver Group ^3^ (*n* = 27)	Overall *p*-Value	Corrected *p*-Value (1 vs. 2)	Corrected *p*-Value (2 vs. 3)	Corrected *p*-Value (1 vs. 3)
Demographics	Age (years)	9.4 ± 2.5	10.2 ± 2.4	11.8 ± 2.3	0.001	0.661	0.037	0.001
	Sex (M:F)	13:18	15:15	19:08	0.086 *	0.527	0.118	0.056
	Height (cm)	139.4 ± 12.7	145.1 ± 10.7	153.9 ± 14.4	<0.001	0.253	0.03	<0.001
	Weight (kg)	42.1 ± 10.5	51.4 ± 13.0	69.8 ± 19.6	<0.001	0.044	<0.001	<0.001
	BMI (kg/m^2^)	21.5 ± 3.4	24.8 ± 3.2	27.9 ± 3.4	<0.001	0.001	0.002	<0.001
	SBP (mmHg)	112.3 ± 10.9	117.3 ± 10.9	120.2 ± 11.9	0.041	0.326	0.999	0.039
	DBP (mmHg)	64.8 ± 10.5	66.5 ± 9.2	68.8 ± 10.3	0.363	0.999	0.999	0.469
	AST (IU/L)	26.8 ± 14.2	25.7 ± 8.7	44.9 ± 26.5	<0.001	0.999	<0.001	0.001
	ALT (IU/L)	17.8 ± 11.9	28.4 ± 16.7	64.4 ± 52.4	<0.001	0.564	<0.001	<0.001
MRI (mild: *n* = 5, moderate to severe: *n* = 15)	Liver longitudinal diameter (cm)		16.0 ± 1.0	18.0 ± 2.3	0.089 **			
	PDFF (%)		12.9 ± 10.4	25.2 ± 10.8	0.04 **			
	T2* (ms)		19.0 ± 3.5	16.5 ± 2.5	0.087 **			
	MRE (kPa)		2.1 ± 0.2	2.0 ± 0.3	0.805 **			
US	ATI median (dB/cm/MHz)	0.54 ± 0.09	0.63 ± 0.08	0.73 ± 0.11	<0.001	0.001	<0.001	<0.001
	ATI IQR	0.09 ± 0.1	0.06 ± 0.03	0.06 ± 0.03	0.125	0.287	0.999	0.201
	ATI IQR/median	0.14 ± 0.07	0.1 ± 0.06	0.08 ± 0.04	0.001	0.028	0.575	0.001
	SWE median (m/s)	1.19 ± 0.07	1.29 ± 0.16	1.26 ± 0.14	0.009	0.019	0.999	0.026
	SWE IQR	0.12 ± 0.06	0.12 ± 0.07	0.13 ± 0.09	0.995	0.999	0.999	0.999
	SWE IQR/median	0.10 ± 0.05	0.09 ± 0.06	0.09 ± 0.06	0.721	0.999	0.999	0.999
	SWE median (kPa)	4.39 ± 0.62	5.1 ± 1.46	5.04 ± 1.44	0.059	0.092	0.999	0.156
	SWD median [(m/s)/kHz]	12.39 ± 1.41	12.45 ± 1.10	12.57 ± 1.26	0.872	0.999	0.999	0.999
	SWD IQR	1.6 ± 0.82	1.76 ± 1.05	1.4 ± 0.77	0.327	0.999	0.999	0.999
	SWD IQR/med	0.13 ± 0.07	0.14 ± 0.08	0.11 ± 0.06	0.302	0.999	0.374	0.999

One-way ANOVA with Bonferroni correction. * Chi-square test. ** independent two-sample *t*-test. US: ultrasonography; BMI: body mass index; SBP: systolic blood pressure; DBP: diastolic blood pressure; AST: aspartate transaminase; ALT: alanine aminotransferase; PDFF: proton density fat fraction; MRE: MR elastography; ATI: attenuation imaging; IQR: interquartile range; SWE: shear-wave elastography; SWD: shear-wave dispersion.

**Table 2 biology-11-01087-t002:** Results of the ordinal logistic regression analysis for the three fatty liver groups (normal, mild, and moderate–severe) according to grayscale US.

Variables		Univariate Analysis			Multivariate Analysis		
		OR	95% CI	*p*-Value	OR	95% CI	*p*-Value
Demographics	Age (years)	0.311	0.137–0.485	<0.001	0.122	−0.163–0.407	0.401
	Sex (male)	0.854	0.065–1.643	0.034	−0.792	−2.120–0.535	0.242
	(female)	(Reference)			(Reference)		
	Height (cm)	0.068	0.035–0.102	<0.001			
	Weight (kg)	0.09	0.056–0.123	<0.001			
	BMI (kg/m^2^)	0.385	0.247–0.524	<0.001	0.453	0.197–0.709	0.001
	SBP (mmHg)	0.048	0.010–0.086	0.013	−0.029	−0.091–0.034	0.364
	DBP (mmHg)	0.032	−0.010–0.074	0.136			
	AST (IU/L)	0.055	0.023–0.088	0.001	0.085	−0.058–0.227	0.246
	ALT (IU/L)	0.06	0.034–0.087	<0.001	0.007	−0.063–0.076	0.852
US	ATI median (dB/cm/MHz)	15.536	10.186–20.886	<0.001	22.09	11.949–32.230	<0.001
	ATI IQR	−10.53	−22.387–1.326	0.082			
	ATI IQR/median	−13.46	−21.190–−5.729	0.001			
	SWE median (m/s)	3.746	0.511–6.980	0.023	0.725	−3.773–5.224	0.752
	SWE IQR	0.099	−4.682–4.879	0.968			
	SWE IQR/median	−2.252	−8.895–4.390	0.506			
	SWE median (kPa)	0.302	−0.045–0.649	0.088			
	SWD median [(m/s)/kHz]	0.089	−0.225–0.402	0.579			
	SWD IQR	−0.157	−0.596–0.282	0.483			
	SWD IQR/median	−2.369	−7.961–3.223	0.406			

US: ultrasonography; OR: odds ratio; CI: confidence interval; BMI: body mass index; SBP: systolic blood pressure; DBP: diastolic blood pressure; AST: aspartate transaminase; ALT: alanine aminotransferase; ATI: attenuation imaging; IQR: interquartile range; SWE: shear-wave elastography; SWD: shear-wave dispersion.

**Table 3 biology-11-01087-t003:** Results of the linear regression analysis for ATI median values in all groups.

Variables		Univariate Analysis			Multivariate Analysis		
		Estimate	Standard Error	*p*-Value	Estimate	Standard Error	*p*-Value
Demographics	Age (years)	0.008	0.005	0.131			
	Sex	−0.044	0.025	0.089			
	Height (cm)	0.001	0.001	0.122			
	Weight (kg)	0.002	0.001	0.006			
	BMI (kg/m^2^)	0.008	0.003	0.013	−0.019	0.009	0.047
	SBP (mmHg)	0.003	0.001	0.011	0.002	0.002	0.314
	DBP (mmHg)	0.002	0.001	0.08			
	AST (IU/L)	0.002	0.001	0.002			
	ALT (IU/L)	0.001	0.001	<0.001	0.001	0.001	0.998
MRI (mild: *n* = 5, moderate to severe: *n* = 15)	Liver longitudinal diameter (cm)	−0.012	0.01	0.235			
	PDFF (%)	0.005	0.002	0.014	0.007	0.005	0.147
	T2* (ms)	−0.019	0.007	0.012	0.009	0.019	0.653
	MRE (kPa)	−0.017	0.098	0.863			
US	SWE median (m/s)	0.067	0.093	0.473			
	SWD median [(m/s)/kHz]	−0.003	0.011	0.765			

BMI: body mass index; SBP: systolic blood pressure; DBP: diastolic blood pressure; AST: aspartate transaminase; ALT: alanine aminotransferase; MRI: magnetic resonance imaging; PDFF: proton density fat fraction; MRE: MR elastography; US: ultrasonography; ATI: attenuation imaging; SWE: shear-wave elastography; SWD: shear-wave dispersion.

**Table 4 biology-11-01087-t004:** Diagnostic performance of median ATI values for staging hepatic steatosis according to the grayscale US and MRI-PDFF.

Reference Modalities	Hepatic Steatosis Groups	Cutoff Value of ATI (dB/cm/MHz)	AUC	95% CI	Sensitivity (%)	Specificity (%)
Grayscale US	normal vs. mild/moderate/severe	0.59	0.853	0.762–0.920	80.7	77.4
	mild vs. moderate/severe	0.69	0.778	0.649–0.878	70.4	83.3
MRI-PDFF	S1 (mild) vs. S2–3 (moderate–severe)	0.69	0.909	0.695–0.990	81.8	88.9

US: ultrasonography; PDFF: proton density fat fraction; ATI: attenuation imaging; AUC: area under the curve; CI: confidence interval.

**Table 5 biology-11-01087-t005:** Comparison of factors related to measurement failure with ATI.

	Success (*n* = 88)	Failure (*n* = 23)	*p*-Value *	OR **	95% CI **	*p*-Value **
Age (years)	10.4 ± 2.6	8.5 ± 1.4	0.065	1.638	1.042–2.576	0.033
Sex	47:41	5:18	0.007	0.242	0.083–0.710	0.01
Height (cm)	145.8 ± 13.8	134.7 ± 9.9	0.069	1.078	1.030–1.129	0.001
Weight (kg)	53.8 ± 18.4	36.4 ± 8.3	<0.001	1.124	1.057–1.196	<0.001
BMI (kg/m^2^)	24.6 ± 4.2	19.9 ± 3.6	0.421	1.361	1.168–1.585	<0.001
Liver echo on US (normal vs. abnormal)	31:57	18:05	<0.001	6.619	2.241–19.552	0.001

* independent two-sample *t*-test or Chi-square test. ** univariate logistic regression result. US: ultrasonography; BMI: body mass index; ATI: attenuation imaging; OR: odds ratio; CI: confidence interval.

## Data Availability

The data that support the findings of this study are available from the corresponding author upon reasonable request.

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
