# Peer review of "Feasibility of Ultrasound Attenuation Imaging for Assessing Pediatric Hepatic Steatosis"

_biology, 2022, doi:10.3390/biology11071087_

Round 1

Reviewer 1 Report

This study presents the performance of ultrasound-based ATI technology for pediatric hepatic steatosis assessment. The study is well written with good science, novelty and good methodology. Some parts of the Introduction section need improvement and the Results section could be enhanced with more figures.

The issues below should be addressed before publication

1. The introduction section lacks of some literature regarding other commercial variants of ATI (e.g. Attenuation and LiSA) and is restricted only to the most popular, CAP. Please include some literature and performances of competitive US attenuation-based techniques. For example, a very recent one by Gatos et al is: doi: 10.1097/RUQ.0000000000000605 

2. Materials and Methods: What is R2 and why a value < 0.8 was set as an exclusion criterium threshold? Should the Authors provide a citation for that?

3. Results: Please explain more why the Bonferoni criterium was used regarding the significance of your results.

4. Please mention explicitly whether there was an ROC analysis performed for optimum cut-off value extraction. The AUCs you mention refer to AUROCs i suppose. Please address.

5. The Results section could be enhanced/enriched visually:

a) By Figures showing the distribution of data per steatosis grade (e.g. Boxplot or Violin plot of ATI values per steatosis grade)

b) By Figures of the ROC plots per steatosis grade from where the AUCs and cut-off values were extracted

Author Response

Dear Editor,

Thank you for your considerate review and suggestions for the revision of our manuscript entitled “Feasibility of ultrasound attenuation imaging for assessing pediatric hepatic steatosis”. We have reviewed the suggestions made by the Reviewers and have done our best to revise the manuscript accordingly. Please find our responses below.

Reviewer(s)' Comments to Author:

Reviewer: 1

This study presents the performance of ultrasound-based ATI technology for pediatric hepatic steatosis assessment. The study is well written with good science, novelty and good methodology. Some parts of the Introduction section need improvement and the Results section could be enhanced with more figures.

→ We are delighted that our findings were found interesting and reviewed in a positive light. We truly appreciate the comments and insightful suggestions made by the Reviewer and have tried to revise the manuscript in accordance with all the recommendations. We hope that the changes will be found satisfactory.

R1-1. The introduction section lacks of some literature regarding other commercial variants of ATI (e.g. Attenuation and LiSA) and is restricted only to the most popular, CAP. Please include some literature and performances of competitive US attenuation-based techniques. For example, a very recent one by Gatos et al is: doi: 10.1097/RUQ.0000000000000605

→ Thank you for your valuable comments. We agree that providing more references on other techniques is required to emphasize the importance of our investigation and to give a fuller picture of which niche ultrasound-based ATI technology will fill. Based on the references given above, we have described other techniques in the manuscript including liver ultrasound attenuation (LiSA).  

R1-2. Materials and Methods: What is R2 and why a value < 0.8 was set as an exclusion criterium threshold? Should the Authors provide a citation for that?

→ Thank you for your comment. The R2 value represents the coefficient of determination and R2 ≥ 0.8 was considered to indicate valid measurements per advice from the vendors. This value is provided along each acquisition to confirm the adequacy of the ROI location and the accuracy of measurement by operators. We added the abovementioned explanation and references for this threshold.

R1-3. Results: Please explain more why the Bonferoni criterium was used regarding the significance of your results.

→ Thank you for your thorough review. In the case of hypothesis testing in which the alternative hypothesis is displayed in the form of "Not H0" like ANOVA, if a p-value is lower than the significance level (alpha=0.05) and the alternative hypothesis is adopted, we perform a post hoc analysis to find the group with the actual difference. When performing a post hoc analysis, the same test is performed several times by pairing each group. With this process, statistically, an increase in type I error occurs, which can cause a bias in which a statistically non-significant result might be judged as significant. Therefore, in our study, we applied the Bonferroni method, the most popular and conservative correction method, to suppress any increases in type I error that may occur from performing multiple tests and then, found the results.

R1-4. Please mention explicitly whether there was an ROC analysis performed for optimum cut-off value extraction. The AUCs you mention refer to AUROCs suppose. Please address.

→ As mentioned in the main text, AUC indicated the area under the ROC curve. In general, a cutoff threshold for AUC is found by inversely calculating the value of the point that is maximized by combining sensitivity and specificity. Therefore, in this study, we calculated the cutoff according to the conventional method mentioned above and assessed prediction performance with this cutoff value. We also presented the results related to the cutoff values in the results (Table 4, AUC, AUC 95% CI, Sensitivity, Specificity). This term was used under the guidance of a statistician, as it was used for many other diagnostic studies that have used the same methods and descriptions. We added a more detailed explanation in the statistical analysis paragraph and we also added ROC curves in response to Comment R1-5.

R1-5. The Results section could be enhanced/enriched visually:

  1. a) By Figures showing the distribution of data per steatosis grade (e.g. Boxplot or Violin plot of ATI values per steatosis grade)

→ Thank you for your idea. We added box-whisker plots for the median ATI values of each hepatic steatosis group in Figure 3.

  1. b) By Figures of the ROC plots per steatosis grade from where the AUCs and cut-off values were extracted

→ Following your advice, we added ROC curves for each hepatic steatosis group in Figure 4. The manuscript was changed accordingly to reflect this new addition.

Reviewer 2 Report

This article presents an important study on assessing the use of ultrasound attenuation imaging (ATI) for diagnosing pediatric hepatic steatosis. It is shown to differentiate between normal liver and several stages of fatty liver. While the results are promising, I have few comments which need to be addressed before accepting the article for publication.

1). How was the ATI performed? Only one line is mentioned in Section 2.3 about ATI acquisition, but the details about the adopted ATI methodology are missing. The authors should add that. This will make the article complete and is also essential to know to analyse and compare other existing ATI methodologies for assessing pediatric hepatic steatosis.

2). Recent years have seen many works on ATI approaches, especially in real-time. In the introduction, the authors should cite the relevant works. An explanation on the choice of the adopted approach by the authors will be beneficial as well.

3). In addition to ATI, other measurements corresponding to SWE, SWD were also obtained. While all these values were used for statistical analysis, the discussion section completely misses them and focuses only on the ATI. Even though the ATI results are the most prominent ones, other values should also be discussed to provide a complete picture to the reader.

4). Of all the variables analysed, height and weight are also significantly different at many instances, but not mentioned anywhere. For instance, in section 3.1 when referring to Table 1; in section 3.2 when referring to Table 2; weight in section 3.3 (table 3); section 3.5 (Table 5). Please complete it.

Minor comments

1.      Line 20 in summary: “… fat deposition in the liver using real-time” should be replaced by “…in real-time”.

2.      The word “code” should be omitted at the end of Section 2.3

3.      Section 2.4: “… that could predicted increased ATI values” should be replaced by “…predict…”.

4.      In Tables 2 and 3, categorise the relevant variables as demographics, MRI and US measurements (as done for Table 1).

5.      Section 3.4: How was the cutoff value of 0.59 dB/cm/MHz determined for ATI differentiation between normal and fatty liver?

6.      Remove the space appearing between two lines of Table 4 caption.

7.      Section 3.5: correct the units of BMI: kg/m2

8.      In conclusions: Add “respectively” at the end of the sentence “… severe fatty liver in children”.

Author Response

Dear Editor,

Thank you for your considerate review and suggestions for the revision of our manuscript entitled “Feasibility of ultrasound attenuation imaging for assessing pediatric hepatic steatosis”. We have reviewed the suggestions made by the Reviewers and have done our best to revise the manuscript accordingly. Please find our responses below.

Reviewer: 2

This article presents an important study on assessing the use of ultrasound attenuation imaging (ATI) for diagnosing pediatric hepatic steatosis. It is shown to differentiate between normal liver and several stages of fatty liver. While the results are promising, I have few comments which need to be addressed before accepting the article for publication.

→ We truly appreciate the Reviewer’s encouraging comments and painstaking attention to detail. We have tried to revise the manuscript according to the Reviewer’s recommendations and we hope that our additional results strengthen our overall conclusions.

R2-1. How was the ATI performed? Only one line is mentioned in Section 2.3 about ATI acquisition, but the details about the adopted ATI methodology are missing. The authors should add that. This will make the article complete and is also essential to know to analyze and compare other existing ATI methodologies for assessing pediatric hepatic steatosis.

→ Throughout the paragraph, we have mentioned the methods needed for ATI acquisition, including procedural details on the operator, machine, transducer, patient preparation, position, ROI placement and measurement numbers. Following the Reviewer’s suggestion, we added an explanation on the principle behind ATI and described the acquisition methods in more depth. In addition, we added references for readers who wish to read up more about the acquisition technique.

R2-2. Recent years have seen many works on ATI approaches, especially in real-time. In the introduction, the authors should cite the relevant works. An explanation on the choice of the adopted approach by the authors will be beneficial as well.

→ This was also a concern raised in Comment R1-1, and we added citations of relevant studies demonstrating the usefulness of ATI to provide a broader picture of why we chose this ATI technique for pediatric hepatic steatosis.

R2-3. In addition to ATI, other measurements corresponding to SWE, SWD were also obtained. While all these values were used for statistical analysis, the discussion section completely misses them and focuses only on the ATI. Even though the ATI results are the most prominent ones, other values should also be discussed to provide a complete picture to the reader.

→ Thank you for your valuable comment. We agreed that we needed to expand more on SWD and we added a more rounded discussion on the other values in this revision.

R2-4. Of all the variables analysed, height and weight are also significantly different at many instances, but not mentioned anywhere. For instance, in section 3.1 when referring to Table 1; in section 3.2 when referring to Table 2; weight in section 3.3 (table 3); section 3.5 (Table 5). Please complete it.

→ Thank you for pointing this out. We agree that mentioning height and weight is important because they were significantly different in the tables. We added and corrected the sentences about height and weight in the Results.

Minor comments

M1. Line 20 in summary: “… fat deposition in the liver using real-time” should be replaced by “…in real-time”.

→ Thank you for your clarification. We corrected the sentence as suggested.

M2. The word “code” should be omitted at the end of Section 2.3

→ We apologize for the oversight. We omitted the word "code” as requested.

M3. Section 2.4: “… that could predicted increased ATI values” should be replaced by “…predict…”.

→ Thank you once again for pointing out this mistake in grammar. We corrected it and also looked through the overall manuscript for other errors we might have overlooked.

M4. In Tables 2 and 3, categorise the relevant variables as demographics, MRI and US measurements (as done for Table 1).

→ Thank you for your valuable comment. We agreed that categorizing the relevant variables would make it much easier for readers to understand the data in the tables. We modified them as suggested.

M5. Section 3.4: How was the cutoff value of 0.59 dB/cm/MHz determined for ATI differentiation between normal and fatty liver?

→ As mentioned in the statistical analysis and Results sections, the cutoff values for diagnostic performance were obtained using AUC analysis. We added another brief mention of this in the Methods section.

M6. Remove the space appearing between two lines of Table 4 caption.

→ Thank you for your explicit comments. We removed the space as suggested.

M7. Section 3.5: correct the units of BMI: kg/m2

→ Thank you for catching this mistake. We corrected the units of BMI accordingly.

M8. In conclusions: Add “respectively” at the end of the sentence “… severe fatty liver in children”.

→ We added "respectively” at the end of the sentence as recommended.

Authors

A1. The authors edited the typo and grammar errors.

We truly appreciate the time and effort taken by the Editor and Reviewers to review our manuscript and we hope that all of the changes will be found satisfactory.
